# Phytochemical Study on Seeds of *Paeonia clusii* subsp. *rhodia*—Antioxidant and Anti-Tyrosinase Properties

**DOI:** 10.3390/ijms24054935

**Published:** 2023-03-03

**Authors:** Vithleem Klontza, Konstantia Graikou, Antigoni Cheilari, Vasilios Kasapis, Christos Ganos, Nektarios Aligiannis, Ioanna Chinou

**Affiliations:** Laboratory of Pharmacognosy and Chemistry of Natural Products, Faculty of Pharmacy, National and Kapodistrian University of Athens, Panepistimiopolis, 15771 Athens, Greece

**Keywords:** Paeoniaceae, *Paeonia clusii* subsp. *rhodia* (Stearn) Tzanoud, seeds, stilbenes, flavonoids, monoterpenic glucosides, total phenolic content, antioxidant activity, anti-tyrosinase assay

## Abstract

In this study, the black fertile (BSs) and the red unfertile seeds (RSs) of the Greek endemic *Paeonia clusii* subsp. *rhodia* (Stearn) Tzanoud were studied for the first time. Nine phenolic derivatives, *trans*-resveratol, *trans*-resveratrol-4′-O-β-d-glucopyranoside, *trans*-*ε*-viniferin, *trans*-gnetin H, luteolin, luteolin 3′-O-β-d-glucoside, luteolin 3′,4′-di-O-β-d-glucopyranoside, and benzoic acid, along with the monoterpene glycoside paeoniflorin, have been isolated and structurally elucidated. Furthermore, 33 metabolites have been identified from BSs through UHPLC-HRMS, including 6 monoterpene glycosides of the paeoniflorin type with the characteristic cage-like terpenic skeleton found only in plants of the genus *Paeonia*, 6 gallic acid derivatives, 10 oligostilbene compounds, and 11 flavonoid derivatives. From the RSs, through HS-SPME and GC-MS, 19 metabolites were identified, among which nopinone, myrtanal, and *cis*-myrtanol have been reported only in peonies’ roots and flowers to date. The total phenolic content of both seed extracts (BS and RS) was extremely high (up to 289.97 mg GAE/g) and, moreover, they showed interesting antioxidative activity and anti-tyrosinase properties. The isolated compounds were also biologically evaluated. Especially in the case of *trans*-gnetin H, the expressed anti-tyrosinase activity was higher than that of kojic acid, which is a well-known whitening agent standard.

## 1. Introduction

*Paeonia* is the sole genus in Paeoniaceae plant family, distributed within Europe and Asia, and is named after Paeon, a student of the Greek father of medicine Asclepius. The plant has been cultivated in China since the Han dynasty, and is known as the “king of flowers” and/or “flowers of richness and honor” [1]. It is known commercially for its ornamental value due to its impressive and often fragrant flowers. Since antiquity, the use of peonies has been reported in traditional medicine for the treatment of epilepsy conditions, as described by Dioscorides in the 1st century A.D. [2]. In Traditional Chinese Medicine, peonies have been used for the treatment of gynecological problems, cramps, and pain and have been used for their antioxidant, anti-inflammatory, analgesic, and antimicrobial properties [3,4,5].

So far, hundreds of secondary metabolites have been isolated from the various peony species, which have been classified into the following chemical categories: monoterpene glycosides, stilbenes, triterpenes, tannins, steroids, flavonoids, and phenols [3]. Paeoniflorin, a water-soluble monoterpene glycoside, has been observed in all peony species examined to date. It has not been found in any other genus beyond that of peonies and is therefore a chemo-taxonomic indicator of the genus, and it is also suspected to be responsible for some of the medicinal properties of the plant [6,7,8]. *Trans*-gnetin H, a resveratrol (3,5,4′-trihydroxystilbene) trimer that is usually found in plants of the Paeoniaceae and Vitaceae botanical families, has shown equally important pharmacological properties, such as antioxidant, cytotoxic antidiabetic, and anti-inflammatory effects, [9,10,11].

In Greece, eight wild Greek *Paeonia* taxa have been recognized and botanically characterized to date, namely, *P. clusii* subsp. *clusii*, *P. clusii* subsp. *rhodia*, *P. mascula* subsp. *mascula*, *P. mascula* subsp. *hellenica*, *P. mascula* subsp. *icarica*, *P. mascula* subsp. *russi*, *P. peregrina*, and *P. parnassica* [12,13].

Until now, only the roots of *P. clusii* Stern subsp. *clusii* and *P. mascula* L. subsp. *hellenica* and *P. parnassica* have been studied phytochemically, mainly based on the traditional uses and reports described in the literature. These subspecies’ in vitro antimicrobial properties, as well as their prophylactic anticonvulsant properties, have been also reported [2,14,15].

Melanogenesis, and, by extension, the pigmentation of the skin, are the most important factors in protecting against UV radiation, which damages the skin. On the contrary, excessive melanin synthesis and accumulation appears in many types of skin disorders, such as melasma, poikiloderma of Civatte, nevi, periorbital hyperpigmentation, cafe-au-lait spots, and freckles, and it is also associated with an increased risk of developing skin cancer [16]. Tyrosinase is a widely distributed enzyme, which is also found in high concentration in melanocytes and is essential for the biosynthesis of melanin. In recent years, natural products have attracted attention for the development of whitening agents as cosmeceuticals [17,18]. Several assays have exhibited that molecules of plant origin and plant extracts present these activities with the Moraceae family being a characteristic example [19,20].

*P. clusii* subsp. *rhodia* (Stearn) Tzanoud. is a Greek endemic species of the Paeoniaceae family that can be found in the mountains of the island of Rhodes, which has characteristic white flowers and large palm-shaped leaves [12]. We report herein a phytochemical study of its seeds, specifically its black fertile seeds (BSs) and red infertile seeds (RSs), which have not been previously studied.

In the present research, four stilbene derivatives of resveratrol (*trans*-resveratrol, *trans*-resveratrol-4′-*O*-*β*-d-glucopyranoside, *trans*-*ε*-viniferin, and *trans*-gnetin H) were isolated and structurally determined, together with three flavonoids in the form of aglycons and/or glycosides (luteolin, luteolin 3′-*O*-*β*-d-glucoside, luteolin 3′,4′-di-*O*-*β*-d-glucopyranoside), one monoterpene glycoside (paeoniflorin), and phenolic benzoic acid from both BSs and RSs of the plant. Furthermore, the metabolite-rich BS extract was subjected to UHPLC-HRMS analysis, in which 33 metabolites were identified.

In parallel, through GC-MS and HS-SPME analyses of the volatiles of RSs, 19 substances were detected. The total phenolic content (TPC) and the antioxidative and anti-tyrosinase activity were evaluated via an enzymatic assay.

## 2. Results

### 2.1. Identification of Secondary Metabolites 

The BSs of *P. clusii* subsp. *rhodia* were studied phytochemically through UHPLC-HRMS analysis, and 33 metabolites were tentatively identified from the methanolic extract (Appendix A), including 6 monoterpene glycosides of the paeoniflorin type with the characteristic cage-like terpenic skeleton found only in plants of the genus *Paeonia*, 6 gallic acid derivatives, 10 oligostilbene compounds, and 11 flavonoid derivatives (Table 1).

The GC-MS analysis of the RS dichloromethane extracts led to the detection of 11 substances, whereas nine substances were detected in the pentane extract (Table 2). Furthermore, the HS-SPME analysis led to the identification of 10 metabolites, and the main metabolites for all of these analyses were myrtanal and nopinone (Table 2).

### 2.2. Isolation of Secondary Metabolites

From the BS methanolic extract, eight secondary metabolites were isolated through the use of chromatographic techniques and structurally elucidated based on spectral data: *Trans*-resveratol [34], *trans*-ε-viniferin [9], *trans*-gnetin H [35], *trans*-resveratrol-4′-*O*-*β*-d-glucopyranoside [36], luteolin [37], luteolin 3′-*O*-*β*-d-glucoside, luteolin 3′,4′-di-*O*-*β*-d-glucopyranoside [38], and paeoniflorin [39]. Moreover, from the RS dichloromethane extract, benzoic acid, which has been previously detected in several peony species, was isolated [35].

### 2.3. Total Phenolic Content (TPC)

The methanolic extracts of the BSs and RSs exhibited high phenolic contents: 204.62 ± 4.0 mg GAE/g extract for the BS extract and 177.59 ± 7.5 GAE/g extract for the RS extract.

### 2.4. DPPH Assay

The antioxidant properties of both seeds’ methanolic extracts and of selected isolated compounds were determined using the DPPH assay and the results are displayed in Table 3 as percentages of DPPH inhibition in three different concentrations. Two-way analysis of variance (ANOVA) was used to evaluate results (*p*-value summary: *p* < 0.0001) and the significance of differences between means was determined by means of Tukey’s post hoc test. Multiple-comparisons differences at *p* < 0.05 were regarded as statistically significant.

### 2.5. Tyrosinase Inhibitory Activity

The tyrosinase inhibitory activity of the studied methanolic extracts of BSs and RSs and of selected isolated compounds is presented in Table 4. One-way ANOVA was used to compare IC50 mean values (*p* < 0.0001) and two-way ANOVA (*p*-value summary: *p* < 0.0001) followed by Tukey’s post hoc test was used to evaluate the significance of differences between means of the inhibition activity percentages. A *p*-value < 0.05 was considered an indicator of statistical significance.

## 3. Discussion

Monoterpenes and monoterpene glucosides are predominant in *Paeonia* species, according to data in the literature [40]. In this study, six monoterpene glycosides were characterised, including paeoniflorin, galloyl-paeoniflorin, dibenzoyl-paeoniflorin, methyl-debenzoyl-paeoniflorin, albiflorin, and one that was not further identified, referred to in Table 1 as a paeoniflorin isomer. Paeoniflorin is commonly found in the Paeoniaceae family and it is considered a chemotaxonomic marker for this family, having been reported in *P. lactiflora*, *P. suffruticosa*, *P. emodi*, *P. osii*, and *P. anomala* [35], as well as in the roots of the Greek species *P. clusii* subsp. *clusii* [14,15].

To date, stilbenes have been isolated only from the seeds of *Paeonia* species and appear to be resveratrol oligomers. In this study, 10 oligostilbene compounds were identified: *trans*- and *cis*- resveratrol, as well as their dimers *trans* and *cis*-*ε*-viniferin and two trimers in the form of *trans*-gnetin H isomers, along with their glucosides (*trans* and *cis*-resveratrol hexosides, *trans* and *cis*-*ε*-viniferin hexosides). Resveratrol and its oligomers have attracted the interest of researchers as potential therapeutic agents for various diseases due to the wide variety of biological and pharmacological activities they have presented [41,42,43,44]. For this reason, extensive research has been undertaken to find natural stilbene agents, which have so far been reported in plant families such as Dipterocarpaceae, Vitaceae, Cyperaceae, Gnetaceae, Fabaceae (Leguminosae), Paeoniaceae, and Apiaceadoraceae [45,46]. In the family Paeoniaceae, they have been found only in the seeds of peony species such as *P. suffruticosa*, *P. lactiflora* [3,9], *P. rockii* [47], and other paeony trees of Chinese origin, and only in *P. officinalis* [36] from Europe.

A number of gallic derivatives from *Paeonia* species have been observed previously. The results of this study indicated that six of them, including gallic acid, methyl gallate, glucogallin, trigalloyl glucose, tetragalloylglucose, and di-*O*-galloyl-*β*-d-glucopyranose, were detected in the studied Greek plant. These compounds have been very recently reported only once in the seed kernels and hulls of *P. lactiflora* Pall. [21]. Furthermore, benzoic acid has been previously detected in several peony species [35].

In addition to the above constituents, 11 flavonoids, including kaempferol and luteolin and their glucosides, as well as two tetrahydroxyflavone hexosides, were present in the black seeds of the studied peony species. Furthermore, in the chemical category of flavonoids, luteolin and its glucosides (luteolin 3′-*O*-*β*-d-glucopyranoside and luteolin 3′,4′-di-*O*-*β*-d-glucopyranoside) were isolated and structurally determined, whereas luteolin-glucopyranosides were isolated for the first time in peonies and in the whole Paeoniaceae family.

Furthermore, many of the compounds which were identified in the non-polar extracts of RS through GC-MS and HS-SPME, such as nopinone and myrtanal, which were the most abundant compounds, as well as *cis*- and *trans*-myrtanol, E-myrtenol, myrtenal, *β*-pinene oxide, camphene, *α*-methylbenzyl alcohol, and phellandral, have been found in previous studies in several peonies, mainly in their roots and flowers [14,48,49,50] but here they were detected for the first time in seeds and especially in RSs (red seeds) from this genus.

Additionally, among the identified metabolites, luteolin and other flavonoid derivatives, such as kaempferol glucoside and di-glucoside, were found to possess significant DPPH radical scavenging activity in the study presented in [51], suggesting that a high flavonoid content (kaempferol- and luteolin-glycosides) is directly related to strong antioxidant properties [52]. In addition, polyphenols such as gallic acid and methyl gallate were previously reported to be more effective in scavenging free-radicals than the known antioxidant compound *α*-tocopherol [53]. Stilbenes such as resveratrol dimers (*trans*- and *cis*-*ε*-viniferin) and trimers (*trans*-gnetin H) exhibited the strongest antioxidant activity among the stilbenes referred to in the literature [10]. Moreover, resveratrol showed moderate activity which was higher than that of its glycosides *trans* and *cis*-resveratroloside, suggesting that its glucosylation decreased its antioxidant activity, probably due to its low affinity to biological lipid membranes [10,54]. Paeoniflorin and monoterpenes with pinane structures have been found not to inhibit free-radical DPPH [53], which was also confirmed by our results. Therefore, the correlation between the synergistic, antagonistic, and antioxidant activities of phenolic substances in peonies should be further examined in the future.

No previous studies have been conducted on the antioxidant activity of *P. clusii* subsp. *rhodia*, but after comparing our results with those presented for other peonies in the literature, we concluded that the activity of this subspecies is significantly higher than that reported for other subspecies. Specifically, Sevim and coworkers [55] reported that seed extracts from seven herbaceous peony species in Turkey exerted low to moderate radical scavenging activity (below 50% at 2000 μg/mL) against DPPH. In another study [56], the authors concluded that the antioxidant capacity of seed coats of *Paeonia* ranged from 66.56 mg/g to 82.85 mg/g (expressed as GAE mg per 1 g DW). Therefore, it seems that the studied red seeds (RSs) from Greek peonies may serve as a good source for extracts with high antioxidant capacity (84.0 ± 2.1 at 100 μg/mL), but further investigation is suggested and required.

Regarding the isolated compounds, luteolin (IC50 = 13.2 ± 3.5 μg/mL) was characterized as a strong antioxidant in relation to gallic acid (IC50 = 4.2 ± 0.1 μg/mL), which is confirmed by the literature [57]. In addition, *trans*-resveratrol showed higher activity than its glycoside and *trans*-*ε*-viniferin, whereas *trans*-gnetin H showed low inhibition, a fact that is also confirmed by the literature [58,59]. Finally, paeoniflorin showed a negative ability to inhibit free-radical DPPH, as was also confirmed in the literature by Picerno et al. [53] and Wu et al. [60], and according to Matsuda et al. [61], the presence of the galloyl group is essential for the radical scavenging effect. Another study comparing the antioxidant effects of oxypaeoniflorin and paeoniflorin reported that oxypaeoniflorin has strong antioxidant potential in comparison to paeoniflorin although they have very similar chemical structures [62]. Therefore, the pinane structure, combined with the lack of a galloyl group and the lack of a hydroxyl group in its aromatic ring, further weaken the antioxidant potential of paeoniflorin as it has no free electrons to contribute.

According to the anti-tyrosinase test results, the BS extract showed good inhibitory activity in all three concentrations (300, 150, and 75 μg/mL) with IC50 = 20.8 ± 1.8 μg/mL, whereas the RS extract showed low inhibitory activity. All three of the tested and isolated secondary metabolites—*trans*-resveratrol-4′-O-β-d-glucopyranoside, *trans*-gnetin H, and *trans*-*ε*-viniferin—inhibited the tyrosinase enzymes by more than 50% at all tested concentrations, with IC50 values of 28.7 ± 6.8 μg/mL, 5.1 ± 2.3 μg/mL, and 3.7 ± 0.1 μg/mL, respectively. *Trans*-gnetin H and *trans*-*ε*-viniferin exerted comparable and/or higher anti-melanogenic potency than the standard compound kojic acid (IC50 = 2.0 ± 0.7 μg/mL) used. These results confirmed previous enzyme assay outcomes reported in the literature [10,36], in which *trans*-ε-viniferin was found to be more potent than kojic acid and ascorbic, whereas resveratrol exhibited a moderate inhibitory activity quite similar to that of arbutin [63].

## 4. Materials and Methods

### 4.1. Plant Material

Fresh black (BSs) and red seeds (RSs) of *P. clusii* subsp. *rhodia* (Stearn) Tzanoud. (Figure 1) were obtained from Prophet Elias mountain (798 m in height) on the island of Rhodes (South Aegean, Greece) in April 2020. The sample was identified botanically by Dr Bazos I., Section of Ecology and Systematics, Department of Biology, National and Kapodistrian University of Athens, Greece. The seeds were naturally dried, ground using a laboratory mill, and stored in darkness at room temperature.

### 4.2. Chemicals and Reagents

Stationary phases for column chromatography (CC) and vacuum liquid chromatography (VLC): silica gel (Kieselgel 60 H Merck), Sephadex LH-20 (25–100 mm, Pharmacia), gradient elution with the solvent mixtures indicated in each case. The solvents used were HPLC-grade and were purchased from Fisher Chemical (Fisher Scientific, Loughborough, Leics, UK). Fractionation was always monitored via TLC: Merck silica gel 60 F254 (0.2 mm layer thickness), Merck RP-18 F254S, and Merck cellulose. For preparative thin-layer chromatography (prepTLC), 60 F254 (Merck) silica gel was used. Detection on TLC plates was enabled using UV light (254 and 366 nm), H_2_SO_4_-vanillin spray reagent on silica gel, and Naturstoff spray reagent on cellulose, followed by heating.

### 4.3. Extraction, Fractionation, and Purification Procedures

#### 4.3.1. Black Seeds of *P. clusii* subsp. *rhodia*

The dried fragmented BSs of *P. clusii* subsp. *rhodia* (16.0 g) were successively extracted in 1 L methanol (MeOH) for 24 h at room temperature to obtain crude methanolic extract (7.5 g). The methanolic extract was subjected to vacuum liquid chromatography (VLC) (gradient elution with cyclohexane/CH_2_Cl_2_ 70:30 to 0:100, CH_2_Cl_2_/EtOAc 80:20 to 0:100, and am EtOAc/MeOH gradient of 99:1 to 70:30) to yield 84 fractions (PB1-PB84). Fractions PB6, PB7, PB8, and PB9, eluted with CH_2_Cl_2_/EtOAc (80:20 to 50:50), were combined (1.4 gr) and further fractionated via CC over silica gel using as eluent mixtures of CH_2_Cl_2_/EtOAc/MeOH (9.7:0.3:0 to 0:7:30) and afforded *trans*-resveratol (12 mg), luteolin (11 mg), *trans*-*ε*-viniferin (57 mg), and *trans*-gnetin H (428 mg). Fractions PB27 to PB36, eluted with CH_2_Cl_2_/EtOAc/MeOH (1:9:0 to 0:9.9:0.1), were combined (717.2 mg) and subjected to CC over silica gel using as eluents mixtures of cyclohexane/EtOAc (50:50 to 0:100) and EtOAc/MeOH (99:1 to 70:30) to afford *trans*-resveratrol-4′-O-*β*-d-glucopyranoside (25.6 mg) and another fraction (231 mg) was further purified through CC with silica gel using CH_2_Cl_2_/MeOH (95:5 to 30:70) as eluent mixtures to yield paeoniflorin (52.6 mg) and luteolin 3′-O-β-d-glucopyranoside (5.3 mg). Combined fractions PB78 to PB85, eluted with EtOAc/MeOH (30:70), were subjected to CC over Sephadex (MeOH) and yielded luteolin 3′,4′-di-O-*β*-d-glucopyranoside (5.6 mg).

#### 4.3.2. Red Seeds of *P. clusii* subsp. *rhodia*

The dried fragmented RSs of *P. clusii* subsp. *rhodia* (6 g) were successively extracted using 40 mL methanol for 24 h at room temperature to obtain crude methanol extract (1.1 g). One part of the extract (700 mg) was subjected to MPLC with gradient elution with mixtures of H_2_O: MeOH of reduced polarity as mobile phase to collect 70 fractions (PR1-PR70). Fractions PR11 and PR12, which were eluted with H_2_O/MeOH (80:20 and 70:30), were combined (36.1 mg) and subjected to preparative TLC with a mixture of CHCl_3_:MeOH at 75:25 to afford paeoniflorin (15.2 mg).

A part of the dried fragmented RSs (500 mg) was extracted with 10 mL of dichloromethane (DCM) and another quantity of 500 mg with 10 mL of pentane C_5_H_12_ using ultrasound for 30 min at 25 °C to yield 7.4 mg and 5.1 mg, respectively. These extracts were further analyzed through GC-MS.

A part of the crude RS DCM extract (25 mg) was subjected to preparative TLC with a mixture of toluene:EtOAc 70:30 *v*/*v* as a mobile phase to afford benzoic acid (2.1 mg).

### 4.4. Analysis of the Volatile Constituents

#### 4.4.1. Headspace Solid-Phase Microextraction (HS-SPME)

The aroma of the *P. clusii* subsp. *rhodia* red seeds was studied through HS-SPME/GC-MS analyses [64]. HS-SPME was performed with Supelco SPME fiber 100 μm PDMS (polydimethylsiloxane coating) and Supelco SPME fiber 75 μm PDMS/DVB (polydimethylsiloxane/divinylbenzene coating) attached to a manual SPME fiber holder (Supelco, Bellefonte, PA, USA). First, the fiber was conditioned in the GC at 250 °C for 30 min and then was inserted into the sample vial. For SPME extraction, 500 mg of the sample in a glass vial (15 mL), closed with a PTFE-coated silicone rubber septum, was used. The temperature in our experiment was set at 60 °C and the vial with the sample was placed on the hotplate for 30 min. After that time, the fiber was exposed to the sample for 30 min at 60 °C and then it was transferred to perform GC-MS analysis. The initial temperature of the column was 55 °C for 2 min with an increase of 5 °C/min until it reached 240 °C, where it remained for 2 min. The total analysis time was 41 min. The substances were identified by comparing the mass spectrum of each substance with those of the Wiley 275 library and the literature.

#### 4.4.2. GC-MS Analysis

The analyses were performed on an Agilent Technologies Gas Chromatography 7820A sysem (Shanghai, China), connected to an Agilent Tecnologies 5977B mass spectrometer (Santa Clara, CA, USA), which worked via EI with an ionization energy of 70 eV. The gas chromatograph was equipped with a split/splitless injector and a 30 m long HP 5MS capillary column with an inner diameter of 0.25 mm and a membrane thickness of 0.25 μm. The temperature in the injection sample was 250 °C and the carrier gas was helium. The following temperature program was used: an initial column temperature of 60 °C was kept constant for 5 min and with an increase of 3 °C/min it reached 280 °C, where it remained for 15 min. The total analysis time was 93 minutes. Identifications were made using the Wiley275 library and bibliographic data.

### 4.5. Nuclear Magnetic Resonance (NMR)

^1^H NMR and 2D-NMR (COSY, HMBC, HSQC) spectra were recorded on a Bruker (Rheinstetten, Germany) DRX 400 (400 MHz) spectroscopy instrument using CD3OD, CDCl3 as a solvent and TMS as an internal standard.

### 4.6. Analysis through UHPLC-HRMS

Ultra-high-performance liquid chromatography was performed employing a Vanquish UHPLC system (Thermo Fisher Scientific, Germering, Germany) equipped with a binary pump, an autosampler, an online vacuum degasser, and a temperature-controlled column compartment. LC-MS-grade acetonitrile (ACN), methanol (MeOH), and formic acid (FA) were purchased from Fisher Scientific (Thermo Fisher Scientific, Leicestershire, UK) and LC-MS water was produced using a Barnstead MicroPure Water Purification System (Thermo Scientific, Germany). An Accucore Vanquish UPLC C18 (2.1 × 50 mm, 1.5 μm) reverse-phase column (Thermo Scientific, Germany) was used for the analysis. High-resolution mass spectrometry was performed on an Orbitrap Exactive Plus mass spectrometer (Thermo Scientific, Germany). Samples were injected at concentrations of 100 ppm diluted in MeOH/H_2_O at 50:50. The mobile phase consisted of solvents A (aqueous 0.1% (*v*/*v*) formic acid) and B (acetonitrile). Different gradient elutions were performed for positive- and negative-ion-mode detection and after optimization of the chromatography the gradient applied was: T = 0 min, 5% B; T = 3 min, 5% B; T = 21 min, 95% B; T = 26 min, 95% B; T = 26.1 min, 5% B; T = 30 min, 5% B. The flow rate was 0.3 mL/min and the injection volume was 5 μL. The column temperature was kept at 40 °C and the sample tray temperature was set at 10 °C. The ionization was performed at HESI, in both positive and negative modes. The conditions for the HRMS for both negative and positive ionization modes were set as follows: capillary temperature, 320 °C; spray voltage, 2.7 kV; S-lense Rf level, 50 V; sheath gas flow, 40 arb. units; aux gas flow, 8 arb. units; aux. gas heater temperature, 50 °C. Analysis was performed using the Fourier transform mass spectrometry mode (FTMS) in the full-scan ion mode, applying a resolution of 70,000, whereas the acquisition of the mass spectra was performed in every case using the centroid mode. The data-dependent acquisition capability was also used at 35,000 resolution, allowing for MS/MS fragmentation of the three most intense ions of every peak exceeding the predefined threshold, applying a 10 s dynamic exclusion window. The normalized collision energy was set at 35. Data acquisition and analysis were completed employing Xcalibur 2.1.

### 4.7. Total Phenolic Content (TPC)

The total phenolic content of the samples was determined by means of the Folin–Ciocalteu method [64]. In a 96-well plate, 25 μL of extracts with different concentrations (4, 2, 1 mg/mL) or standard solutions of gallic acid (2.5, 5, 10, 12.5, 20, 25, 40, 50, 80, 100 g/mL), both diluted in DMSO, were added to 125 μL of a Folin–Ciocalteu solution (10%), followed by the addition of 100 μL of 7.5% sodium carbonate. The plate was incubated for 30 min in darkness at room temperature. The absorbance at 765 nm was measured using a TECAN Infinite m200 PRO multimode reader (Tecan Group, Männedorf, Switzerland). All measurements were performed in triplicate, the mean values were plotted on a gallic acid calibration curve, and the total phenolic content was expressed as mg equivalents to gallic acid (GAE) per gram of dry extract.

### 4.8. DPPH (2,2-DiPhenyl-1-PicrylHydrazyl) Assay 

The antioxidant activity of the samples was evaluated based on DPPH (1,1-diphenyl-2-picrylhydrazyl) radical scavenging activity according to the literature [65]. For the DPPH assay, the methanol extract and the isolated compounds (1, 2, 3, 4, 5, and 6) (concentration of stock solution: 4 mg/mL) were prepared using dimethylsulfoxide (DMSO) as a solvent. Ten (10) μL of each sample were mixed with 190 μL of DPPH solution (12.4 mg/100 mL in ethanol) in a 96-well plate and then subsequently incubated, at room temperature, for 30 min in darkness. Finally, the absorbance was measured at 517 nm using an Infinite M200 Pro TECAN photometer (Tecan Group, Männedorf, Switzerland). All evaluations were performed in triplicates, whereas gallic acid was used as positive control (IC50 = 4.2 ± 0.1 μg/mL). The percentage inhibition of the DPPH radical for each dilution was calculated using the following formula: % inhibition = [(A − B) − (C−D)]/(A − B) × 100, where A: control (w/o sample), B: blank (w/o sample, w/o DPPH), C: sample, D: blank sample (w/o DPPH). All the samples were tested at final concentrations of 200 μg/mL, 100 μg/mL, and 50 μg/mL, whereas for the most active of them (DPPH inh. >70% at concentration C = 50 μg/mL), the IC50 value was calculated.

### 4.9. Tyrosinase Inhibition 

The methanolic extract, as well as compounds 3, 4, and 5, were investigated for their ability to inhibit the oxidation of L-DOPA (L-3,4-dihydroxyphenylalanine) to dopaquinone and subsequently to dopachrome by the enzyme tyrosinase [19]. The fractions were initially dissolved in DMSO (10 mg/mL) and subsequently diluted in the proper concentration in phosphate buffer, 1/15 M (NaH_2_PO_4_/Na_2_HPO_4_), pH 6.8. Final concentrations of DMSO in the well did not exceeded 3%. In 96-well plates, 80 μL of phosphate buffer 1/15 M (NaH_2_PO_4_/Na_2_HPO_4_), pH 6.8, 40 μL of sample in the same buffer and 40 μL mushroom tyrosinase (92 Units/mL) in the same buffer, were mixed. The contents of each well were incubated for 10 min at 25 °C, before 40 μL of 2.5 mM L-DOPA were added into the same buffer. After incubation at 25 °C for 5 min, the absorbance at 475 nm of each well was measured. Samples were evaluated at 75 μg/mL, 150 μg/mL, and 300 μg/mL, in triplicate, and blank samples for every fraction were also measured, whereas kojic acid was used as a positive control. The percentage inhibition of tyrosinase activity was calculated using the following equation: [(A − B) − (C − D)]/(A − B) × 100, where A: control (w/o sample), B: blank (w/o sample, w/o tyrosinase), C: sample, D: blank sample (w/o tyrosinase).

### 4.10. Statistical Analysis

All in vitro tests were performed in triplicate and values are expressed as mean values ± S.D. of the three independent experiments, calculated in Excel. Then, statistical analysis was performed on the means of the triplicates using GraphPad Prism Version 8.3.0, applying one-way ordinary analysis of variance (ANOVA) for tyrosinase IC50 and two-way ANOVA for the rest of the experiments.

## 5. Conclusions

The black fertile seeds (BSs) and the red unfertile seeds (RSs) of the Greek endemic *Paeonia clusii* subsp. *rhodia* were studied and nine (9) phenolic derivatives were isolated, whereas thirty-three (33) metabolites were identified from BSs through UHPLC-HRMS. The total phenolic content of BSs was extremely high, as well as the anti-tyrosinase activity. The high antioxidant activity expressed by the extracts could suggest that the studied extract could be useful for the treatment of the effects of oxidative stress, which can lead to several skin disorders. In addition to presenting the chemical analysis of Greek Paeonia seeds and their abundant resveratrol derivatives, in this study we have also provided new ideas for the comprehensive development and application of *Paeonia* seeds in the field of skincare, such as dermatology and cosmetology. Due to the increasing demand for natural ingredients from peony species, quantitative methods of analysis for stilbenes, as well as further studies towards dermal absorption, metabolism, and safety of use, will be required for the potential application of such extracts and components in the industry of “eco cosmetics”. Furthermore, the quantification of stilbene derivatives in seeds of wild-growing peony species in Greece will be planned in the near future.

## Figures and Tables

**Figure 1 ijms-24-04935-f001:**
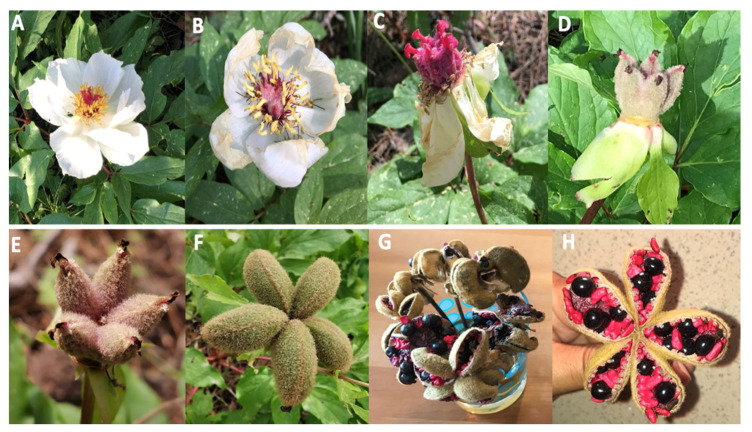
The cycle of flowers (**A**–**G**) to full seed maturation (**H**). Photographs from V. Kasapis, April 2019.

**Table 1 ijms-24-04935-t001:** Secondary metabolites of the BS methanolic extract of *P. clusii* subsp. *Rhodia*, determined using UHPLC-HRMS in negative mode.

No.	Rt (min)	Compound	Molecular Formula	*m/z*	MS/MS Fragment Mass	Literature
1	0.39	bis-hexoses	C_12_H_22_O_11_	341.1092	89, 59	-
2	0.46	citric acid	C_6_H_8_O_7_	191.0192	85, 127, 111, 173, 150	[21]
3	0.51	desbenzoyl-paeoniflorin	C_16_H_24_O_10_	375.1301	165, 345, 89	[21,22]
4	0.57	glucogallin	C_13_H_16_O_10_	331.0676	169, 211, 151, 271	[21,22]
5	1.65	methyl gallate	C_8_H_8_O_5_	183.029	183, 168	[21,23]
6	1.84	di-galloylglucoside	C_20_H_20_O_14_	483.0789	169	[24]
7	4.80	paeoniflorin isomer	C_23_H_28_O_11_	479.1559	121, 165	[21,25]
8	4.88	*O*-methyldesbenzoyl-paeoniflorin	C_17_H_26_O_10_	435.1299	227, 389	[21,25]
9	4.94	trigalloyl glucose	C_8_H_8_O_2_	635.0906	169, 313, 465	[24,26]
10	5.12	*cis*-resveratrol hexoside	C_20_H_22_O_8_	389.1248	227, 185	[21,27]
11	5.22–5.29	Paeoniflorin */albiflorin	C_23_H_28_O_11_	479.1559525.1618	121, 165	[21,28]
12	6.00	kaempferol dihexoside	C_30_H_25_O_14_	609.1472	285, 447	[21,29]
13	6.06	tetragalloyl glucose	C_8_H_8_O_2_	787.1014	169, 465, 313	[24,25,26]
14	6.15	tetrahydroxyflavone hexoside	C_21_H_20_O_11_	447.0936	285	[21,30]
15	6.16	luteolin 3′,4′-di-*O*-*β*-d-glucopyranoside *	C_27_H_30_O_16_	609.1462	285, 447	[21]
16	6.35	tetrahydroxyflavone hexoside	C_21_H_20_O_11_	447.0933	285	[21,30]
17	6.43	galloyl paeoniflorin isomer	C_30_H_32_O_11_	631.1676	169, 313, 211, 271, 121, 399	[25,31]
18	6.72	*trans*-resveratrol-4′-*O*-*β*-*d*-glucopyranoside *	C_20_H_22_O_8_	389.1249	227	[27]
19	6.85	kaempferol hexoside	C_21_H_20_O_11_	447.0938	285	[29]
20	7.13	*cis*-resveratrol	C_14_H_12_O_3_	227.0712	185, 143, 183	[21,27]
21	7.15	luteolin-3′-*O*-*β*-*d*-glucopyranoside *	C_21_H_20_O_11_	447.0938	285	[21]
22	7.87	kaempferol	C_15_H_10_O_6_	285.0406	151, 175	[21]
23	7.87	*cis*-*ε*-viniferin hexoside	C_34_H_32_O_11_	615.1873	453, 347, 359, 333, 227	[21]
24	7.88	kaempferol arabinoside	C_20_H_18_O_10_	417.0833	285	[24]
25	8.00	luteolin *	C_15_H_10_O_6_	285.0408	151, 175	[21]
26	8.15	*trans*-resveratrol *	C_14_H_12_O_3_	227.0711	185, 143, 183	[21,27]
27	8.44	*trans*-*ε*-viniferin hexoside	C_34_H_32_O_11_	615.1873	453, 347, 359, 333, 227	[21]
28	8.49	kaempferol arabinoside	C_20_H_18_O_10_	417.0833	285	[24]
29	8.75	*cis*-*ε*-viniferin	C_28_H_22_O_6_	453.1349	347, 225, 93, 411	[32]
30	8.75	kaempferol arabinoside	C_20_H_18_O_10_	417.0833	285	[24]
31	9.15	*trans*-*ε*-viniferin *	C_28_H_22_O_6_	453.1347	347, 225, 93, 411	[32]
32	9.20	*cis*-gnetin H	C_42_H_32_O_9_	679.1984	93, 491, 478, 449, 357, 225, 585	[27,33]
33	9.63	*trans*-gnetin H *	C_42_H_32_O_9_	679.1981	93, 345, 225, 491, 449, 357, 585	[27,33]

* isolated compounds.

**Table 2 ijms-24-04935-t002:** Tentative identification of compounds of RS extracts from *Paeonia clusii* subsp. *rhodia* via GC-MS (% area).

Compound	HS-SPME	GS-MS of DCM Extract	GS-MS of Pentane Extract
nopinone	20.64 ± 0.17	15.65 ± 0.76	17.54 ± 0.33
*p*-menth-3-en-9-ol	5.96 ± 0.06	-	3.32 ± 0.24
camphene	4.76 ± 0.15	-	5.33 ± 0.26
*a*-methyl- benzenemethanol	-	8.60 ± 0.65	17.92 ± 0.28
myrtanal	32.67 ± 0.38	14.14 ± 0.30	18.68 ± 0.47
*β*-pinene oxide	3.86 ± 0.24	6.57 ± 0.36	-
myrtenal	3.75 ± 0.08	-	-
1-ethanol-2,2,4-trimethyl-3-cyclopentene	4.58 ± 0.16	-	-
*cis*-myrtanol	3.73 ± 0.13	3.79 ± 0.14	2.18 ± 0.25
*trans*-myrtanol	9.80 ± 0.09	-	-
unknown	5.94 ± 0.15	21.54 ± 0.22	25.41 ± 0.90
perillyl alcohol	4.31 ± 0.11	-	-
E-myrtenol	-	1.81 ± 0.11	-
phellandral	-	2.77 ± 0.07	3.85 ± 0.20
5,7-dimethyl-1,6-octadiene	-	4.15 ± 0.29	2.49 ± 0.13
2,4-decadienal	-	2.27 ± 0.06	-
unknown	-	4.50 ± 0.30	-
methyl-3,4-dimethyl benzoate	-	-	2.43 ± 0.25
unknown	-	10.86 ± 0.29	-

**Table 3 ijms-24-04935-t003:** DPPH inhibition by the BS and RS extracts and isolated compounds at different concentrations.

Studied Extracts/Compounds	% DPPH Inhibition			IC50
	200 μg/mL	100 μg/mL	50 μg/mL	μg/mL
BS methanolic extract	37.3 ± 0.6 ^Aa^	19.9 ± 1.8 ^Ba^	0 ^Ca^	-
RS methanolic extract	95.5 ± 0.2 ^Ab^	84.0 ± 2.1 ^Bb^	51.7 ± 2.0 ^Cb^	-
*trans*-resveratrol-4′-*O*-*β*-d-glucopyranoside	39.4 ± 2.9 ^Aa^	29.3 ± 1.5 ^Bc^	20.6 ± 0.8 ^Cc^	-
*trans*-*ε*-viniferin	88.5 ± 0.1 ^Ac^	60.8 ± 1.2 ^Bd^	28.9 ± 5.7 ^Cd^	-
*trans*-gnetin H	39.9 ± 0.9 ^Aa^	24.2 ± 2.5 ^Ba^	16.0 ± 1.0 ^Ce^	-
*trans*-resveratrol	82.1 ± 0.4 ^Ad^	60.5 ± 2.1 ^Bd^	45.0 ± 2.5 ^Cf^	-
luteolin	89.3 ± 0.4 ^Ac^	90.1± 0.6 ^Ae^	92.4 ± 0.1 ^Ag^	13.2 ± 3.5
paeoniflorin	0 ^Ae^	0 ^Af^	0 ^Aa^	-
gallic acid	-	96.3 ^Ag^	95.5 ^Ag^	4.2 ± 0.1

Values expressed are means ± S.D. of three parallel measurements. Superscripts A to C and a to g denote significant differences within rows and columns, respectively, and multiple comparison analysis resulted in statistical significance levels of *p* < 0.05.

**Table 4 ijms-24-04935-t004:** Tyrosinase inhibitory activity of BS and RS extracts and isolated compounds at different concentrations.

Studied Extracts/Compounds	% Tyrosinase Inhibition			IC50
	300 μg/mL	150 μg/mL	75 μg/mL	μg/mL
BS methanolic extract	93.8 ± 1.2 ^Aa^	90.7 ± 2.8 ^Aa^	78.1 ± 2.4 ^Ba^	20.8 ± 1.8
RS methanolic extract	45.9 ± 7.7 ^Ab^	39.2 ± 11.9 ^Ab^	31.9 ± 2.3 ^Bb^	-
*trans*-resveratrol-4′-*O*-*β*-d-glucopyranoside	78.2 ± 2.1 ^Ac^	84.6 ± 1.6 ^Aac^	60.4 ± 2.2 ^Bc^	28.7 ± 6.8
*trans*-*ε*-viniferin	99.1 ± 1.3 ^Aa^	90.7 ± 2.1 ^Bac^	89.6 ± 1.0 ^Bd^	5.1 ± 2.3
*trans*-gnetin H	99.1 ± 1.6 ^Aa^	98.4 ± 1.6 ^Aa,ac^	97.1 ± 0.2 ^Ad^	3.7 ± 0.1
kojic acid	-	96.0 ± 1.5 ^Aa,ac^	95.1 ± 0.8 ^Ad^	2.0 ± 0.7

Values expressed are means ± S.D. of three parallel measurements. Superscripts A to C and a to d denote significant differences within rows and columns, respectively, and multiple comparison analysis resulted in statistical significance levels of *p* < 0.05.

## Data Availability

The data that support the findings of this study are available from the corresponding author upon reasonable request.

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
