# Peer review of "Phytochemical Study on Seeds of Paeonia clusii subsp. rhodia—Antioxidant and Anti-Tyrosinase Properties"

_ijms, 2023, doi:10.3390/ijms24054935_

Round 1

Reviewer 1 Report

The article submited by Vithleem Klontza and cols. present for the first time the phytochemical study on seeds of Paeonia clusii subsp. Rhodia. Although another subsp. have been evaluated previously, it is the first time for this subsp. Introduction is complete  and well-conducted. In the subsequent section comments are listed:

1.       The results of the investigation do not directly reflect what is stated in the title. Further studies are needed to establish that the plant has skin benefits, or the title should be changed to "Phytochemical study on seeds of Paeonia clusii subsp. rhodia. Unlocking potential skin benefits (antioxidant and whitening properties), since the bioactivities tested were not tested in skin cultured cells or skin directly.

2.       In line 74 the scientific name of the plan is not in italics.

3.       Some retention times are so closed.  I suggest to include the chromatogram in Results section, in order to complement the presented results and readers can visualize the compound profile in each analysis were aplicable.

4.       In table 1 does not specify the meaning of (*).

5.       Line 79. In the table title please add the type of extract analyzed.

6.       Line 85. Change the table title for something like: “Tentative identification of…..”

7.       For both Table 3 and 4: include the type of extract and “at different concentrations” somewhere in the title. Additionally, a statistical test is suggested to establish significant differences. Caption of tables are missing, where include details such as “Data are presented as mean ± SD/error of three individual determinations.

8.       Lines 165-167, the study presented does not indicate the value of DPPH at the concentration 66.56-82.85 mg/g, Or it refers to the IC50??

9.       Line 169 change capacity to “capacity”

10.   Lines 174-175 additional discussion for the negative ability of the paeoniflorin will be valuable

11.   In Materials and Methods section. The subsection “General experimental procedures” is not entirely clear. I suggest to divide the information and include an additional subsection of “chemicals” to describe all the chemicals used in the experiments, and the fractionation of extract include in 4.3 . If section 4.3 become so extensive, maybe will be necessary to subdivide this section, to make it clearer.  Include an Statical analysis of data section.

12.   In lines 258 and 259, details about the gas chromatography and mass spectrometer are missing: city, state and country

13. Some references does not fit with the MDPI required format. Some of them are marked with highligh in text.

Author Response

The article submited by Vithleem Klontza and cols. present for the first time the phytochemical study on seeds of Paeonia clusii subsp. Rhodia. Although another subsp. have been evaluated previously, it is the first time for this subsp. Introduction is complete and well-conducted. In the subsequent section comments are listed:

  1. The results of the investigation do not directly reflect what is stated in the title. Further studies are needed to establish that the plant has skin benefits, or the title should be changed to "Phytochemical study on seeds of Paeonia clusii subsp. rhodia. Unlocking potential skin benefits (antioxidant and whitening properties), since the bioactivities tested were not tested in skin cultured cells or skin directly.
  • The title has been changed in order to reflect to the content of the manuscript The new title is “Phytochemical study on seeds of Paeonia clusii rhodia. Antioxidant and Anti-tyrosinase properties”
  1. In line 74 the scientific name of the plan is not in italics.
  • the plant name has been corrected in italics
  1. Some retention times are so closed.  I suggest to include the chromatogram in Results section, in order to complement the presented results and readers can visualize the compound profile in each analysis were aplicable.
  • Τhe chromatogram of methanolic extract has been inserted in the Supplementary File as FS1.
  1. In table 1 does not specify the meaning of (*).
  • the meaning of (*) as “isolated compounds” has been inserted at the end of Table 1
  1. Line 79. In the table title please add the type of extract analyzed.
  • the type of studied extract (methanol) has been added in the title of the Table1
  1. Line 85. Change the table title for something like: “Tentative identification of…..”
  • the title of Table 2 has been changed according to the suggestion
  1. For both Table 3 and 4: include the type of extract and “at different concentrations” somewhere in the title. Additionally, a statistical test is suggested to establish significant differences. Caption of tables are missing, where include details such as “Data are presented as mean ± SD/error of three individual determinations.
  • in both Tables 3 and 4 : the type of extract (methanol) has been added on the first two lines, the explanation at different concentrations” has been added in both titles, the explanation “Values expressed are means ±D. of three parallel measurements” has been added at the end of each table.

Moreover, analysis of variance has been performed and has been added in the relevant section.

  1. Lines 165-167, the study presented does not indicate the value of DPPH at the concentration 66.56-82.85 mg/g, Or it refers to the IC50??
  • In ref [50] the authors express the results as GAE mg per 1 g DW and is mentioned that “the DPPH assay showed that the antioxidant capacity of all samples of subgenus Paeonia was generally higher than that of subgenus Moutan. The content of subgenus Paeonia ranged from 66.56 mg/g (P19) to 82.85 mg/g (P13), while that of subgenus Moutan ranged from 27.81 mg/g (P49) to 67.58 mg/g (P24).” We added “(expressed as GAE mg per 1 g DW)” in parenthesis to clarify
  1. Line 169 change capacity to “capacity”
  • it has been corrected accordingly
  1. Lines 174-175 additional discussion for the negative ability of the paeoniflorin will be valuable
  • An additional paragraph has been added regarding the negative ability of paeoniflorin as well as some extra references on this topic.
  1. In Materials and Methods section. The subsection “General experimental procedures” is not entirely clear. I suggest to divide the information and include an additional subsection of “chemicals” to describe all the chemicals used in the experiments, and the fractionation of extract include in 4.3 . If section 4.3 become so extensive, maybe will be necessary to subdivide this section, to make it clearer.  Include an Statical analysis of data section.
  • The subsection of Material and Methods have been changed. A subsection entitled Chemicals and reagents has been inserted and the fractionation of extracts has been divided in two parts (black seeds and red seeds). Also a subsection entitled Statical analysis has been inserted.
  1. In lines 258 and 259, details about the gas chromatography and mass spectrometer are missing: city, state and country
  • the information regarding city and country of the instruments have been added
  1. Some references does not fit with the MDPI required format. Some of them are marked with highligh in text.
  • All the references have been corrected according to MDPI format.

Reviewer 2 Report

The manuscript in titled (Phytochemical study on seeds of Paeonia clusii subsp. rhodia. Unlocking skin benefits (antioxidant and whitening properties) The topic is interesting however there are some suggestions to improve the manuscript , please use the suggestions in the attached file:

1- In the abstract (line 16 and 19) the apparatus used wrote by abbreviation (Never start with abbreviation)

2- In introduction line 49 (move the reference 12 to the end of sentence)

3- In line 59 (Add reference)

4- There are sentences repeated many times like (Nine phenolic derivatives: trans-12 resveratol, trans-resveratrol-4'-O-β-D-glucopyranoside, trans-ε-viniferin, trans-gnetin H, luteolin, 13 luteolin 3'-O-β-D-glucoside, luteolin 3',4'-di-O-β-D-glucopyranoside, benzoic acid along with mon-14 oterpene glycoside paeoniflorin, have been isolated and structurally elucidated) and for the first time, Its not good in the scientific writing, kindly modify that.

5- In line 69 delete (through Folin-Ciocalteu method,) from her its only in material and method 

6- Please use italics in the scientific name in all the manuscript

7- In line 122 the reference wrote (36-39) not like journal format

8- The correlation between phenolic compounds and antioxidant activity results must be examined in this manuscript and also authors should analyze the antioxidant activity by other methods not only DPPH method its not enough to confirm the results

9- There are no any statistical analysis in the study or any correlations study to confirm the results and showed the significance of the results

10- In line 170, the IC50 results should transfer to antioxidant activity table

11- In line 177, the IC50 results should transfer to anti- tyrosinase activity table

12- The reported results in this discussion section are discussed poorly; please deepen the analysis of the obtained results with updated references

13- In material and methods section, add the city and country of all apparatus used

Author Response

The manuscript in titled (Phytochemical study on seeds of Paeonia clusii subsp. rhodia. Unlocking skin benefits (antioxidant and whitening properties) The topic is interesting however there are some suggestions to improve the manuscript , please use the suggestions in the attached file:

  • In the abstract (line 16 and 19) the apparatus used wrote by abbreviation (Never start with abbreviation)
  • the abbreviations were used in order to keep the correct word limit for the abstract. An abbreviation table has been created to provide all the appropriate info.
  • In introduction line 49 (move the reference 12 to the end of sentence)
  • the reference has been moved to the end of the sentence
  • In line 59 (Add reference)
  • A new reference regarding the botanic characteristics of the plant has been added (ref 13).
  • There are sentences repeated many times like (Nine phenolic derivatives: trans-12 resveratol, trans-resveratrol-4'-O-β-D-glucopyranoside, trans-ε-viniferin, trans-gnetin H, luteolin, 13 luteolin 3'-O-β-D-glucoside, luteolin 3',4'-di-O-β-D-glucopyranoside, benzoic acid along with mon-14 oterpene glycoside paeoniflorin, have been isolated and structurally elucidated) and for the first time, Its not good in the scientific writing, kindly modify that.
  • the appropriate changes have been made to the manuscript in order to avoid repetitions
  • In line 69 delete (through Folin-Ciocalteu method,) from her its only in material and method 
  • the last sentence of the Introduction part has been corrected according to the suggestion
  • Please use italics in the scientific name in all the manuscript
  • italics have been used in all scientific names throughout the manuscript
  • In line 122 the reference wrote (36-39)not like journal format
  • the format has been corrected [36-39]
  • The correlation between phenolic compounds and antioxidant activity results must be examined in this manuscript and also authors should analyze the antioxidant activity by other methods not only DPPH method its not enough to confirm the results
  • The main scope in this investigation was to study the phytochemical profile of the not previously studied seeds mainly towards their phenolic and stilbene oligomers profile rather than to re-examine their already well known antioxidant activity (through enzymatic in vivo and in vitro models). We tried DPPH assay in order to confirm their antioxidant activity depending on their TPC which showed linear relation between the two methods. For the authors (all of us from phytochemical background) DPPH is a horse work easy in our daily studies to have a first proof for further pharmacological studies towards more complicated techniques
  • There are no any statistical analysis in the study or any correlations study to confirm the results and showed the significance of the results
  • Analysis of variance has been performed and added in relevant sections.
  • In line 170, the IC50 results should transfer to antioxidant activity table
  • In line 177, the IC50 results should transfer to anti- tyrosinase activity table
  • the IC50 results have been inserted in both tables 3 and 4
  • The reported results in this discussion section are discussed poorly; please deepen the analysis of the obtained results with updated references
  • Several publications were added in this part (hope not to many, indeed) which to the views of the authors support the choice of this scientific study as well as the importance of the received results, as stilbenes are characterized by a well-known wide-ranging health promoting properties
  • In material and methods section, add the city and country of all apparatus used
  • All the details regarding the apparatus have been added

Reviewer 3 Report

Review report

Phytochemical study on seeds of Paeonia clusii subsp. rhodia Unlocking skin benefits (antioxidant and whitening properties)

The paper entitled: "Phytochemical study on seeds of Paeonia clusii subsp. rhodia. Unlocking skin benefits (antioxidant and whitening properties)", submitted by Klontza et al. is an interesting paper that investigates for the first time the phytochemical composition of Paeonia clusii subsp. rhodia black fertile (BS) and the red unfertile seeds (RS), as well as the antioxidant and the anti-tyrosinase activities related to the plant’s seeds.

·         From a morphological point of view, is there any remarkable differences between the two type of P. clusii subsp. Rhodia seeds. Include some picture (with a scale).

·         Why the authors investigated the phytochemical profile of Fresh black (BS) of P. clusii subsp. Rhodia with the UHPLC-HRMS while for the red seeds (RS), was investigated using GC-MS and HS-SPME?

·         Add the chromatograms of the identified compounds either to the manuscript or as a supplementary file.

·         For the GC-MS results, please add the retention times, RT (min), for each compound to Table 2.

·         The authors must include a paragraph about the anti-tyrosinase activity in natural compounds, and their relevancy as bio-cosmetic ingredients.

General comments:

1.     Abbreviate the words at the start then abbreviate them. The compounds’ names must be in the correct form (e.i. p-menth-3-en-9-ol, with p in Italics – same for the O-linked glycosidic bonds), then utilize them throughout the manuscript, e.i. trans-resveratrol-4'-O-β-D-glucopyranoside (same thing for cis-trans isomerism)).

2.      Is it paeonies, or peony species, are the two terms refer to the same thing?

3.      Avoid formatting mistakes.

Author Response

The paper entitled: "Phytochemical study on seeds of Paeonia clusii subsp. rhodia. Unlocking skin benefits (antioxidant and whitening properties)", submitted by Klontza et al. is an interesting paper that investigates for the first time the phytochemical composition of Paeonia clusii subsp. rhodia black fertile (BS) and the red unfertile seeds (RS), as well as the antioxidant and the anti-tyrosinase activities related to the plant’s seeds.

  • From a morphological point of view, is there any remarkable differences between the two type of P. clusii subsp. Rhodia seeds. Include some picture (with a scale).
  • pictures from the whole cycle of maturation of both black fertile (BS) and red unfertile seeds (RS) have been added in the Material and Methods
  • Why the authors investigated the phytochemical profile of Fresh black (BS) of P. clusiisubsp. Rhodia with the UHPLC-HRMS while for the red seeds (RS), was investigated using GC-MS and HS-SPME?
  • The RS were poor in phenolic compounds but they were odorant and showed a very interesting volatile profile that’s why we as authors studied them through GC techniques. On the other hand BS were a rich source of phenolics (TPC assay) and the authors decided to study them through CC, VLC and investigate the resulting fractions through UHPLC-HRMS in order to identify minor phenolic constituents beside the ones isolated.
  • Add the chromatograms of the identified compounds either to the manuscript or as a supplementary file.
  • the chromatogram of the fractions have been inserted in the Supplementary file.
  • For the GC-MS results, please add the retention times, RT (min), for each compound to Table 2.
  • the RT for the compounds are different as we studied pentane and dichloromethane extract through GCMS but also the sample via Headspace solid-phase microextraction through a different method in the GC analysis
  • The authors must include a paragraph about the anti-tyrosinase activity in natural compounds, and their relevancy as bio-cosmetic ingredients.
  • A paragraph has been added in the Introduction part.

General comments:

  1. Abbreviate the words at the start then abbreviate them. The compounds’ names must be in the correct form (e.i. p-menth-3-en-9-ol, with p in Italics – same for the O-linked glycosidic bonds), then utilize them throughout the manuscript, e.i. trans-resveratrol-4'-O-β-D-glucopyranoside (same thing for cis-trans isomerism)).
  • A list with all abbreviations have been inserted in the manuscript and all the suggested corrections regarding the compounds’ names have been made throughout the paper.

  1. Is it paeonies, or peony species, are the two terms refer to the same thing?
  • the term paeonies in the Conclusion part has been replaced by peony species
  1. Avoid formatting mistakes.
  • formatting mistakes have been corrected throughout the manuscript and the Reference part

Reviewer 4 Report

In my opinion, the manuscript presents preliminary results rather than having the form of a full-fledged paper, includes too few interesting results. For unknown reasons, the authors subjected only one BS extract to LC-MS analysis; and only the RS extract to GC-MS analysis. Biological activity studies include only 2 fairly simple in vitro tests.
Other more specific comments are included in the attached pdf document.
I advise rejection of the manuscript.

Author Response

  • It has to clarify that in this scientific work the seeds of a not studied plant as well as a non studied part (seeds) of this endemic Paeonia have been investigated. It is known that the seeds of this genus are a very rich source of stilbenoids, which are approved bioactive secondary metabolites. Out of China no such studies on seeds have been performed. So the most important approach was to search the seeds and investigate the existence of such metabolites. Both the Black seeds as well as the Red seeds were searched following the classic phytochemical way of fractionations and isolations. It can be seen from the results that the Black seeds were rich containing many different kinds of metabolites, while from the RS only a couple of metabolites were isolated and structurally determined. That is the reason that the BS extract has been further investigated through LC-MS. Contrary, the RS were fragrant and were subjected to GC-MS analyses, while the BS which as a matter of fact are the “ripped” ones did not have any odor, which as a matter of fact proved by the preliminary analysis (not introduced in the text) showing that contained only hydrocarbons and no essential oils’ metabolites. That is the reason that only the BS GC-MS analyses were introduced in the manuscript. The authors were of the opinion that only the interesting analytical results would be included thereof. Furthermore, the main scope in this study was the phytochemical profile of the not previously studied seeds mainly towards their phenolic and stilbene oligomers profile rather than to re-examine their already well known antioxidant activity (through enzymatic in vivo and in vitro models). We tried DPPH assay in order to confirm their antioxidant activity depending on their TPC which showed linear relation between the two methods. For the authors (all of us from phytochemical background) DPPH is a horse work easy in our daily studies to have a first proof for further pharmacological studies.

Round 2

Reviewer 2 Report

The authors made my suggestion and its fit to publish

Author Response

thank you for your consideration and your suggestions which made our manuscript better

Reviewer 4 Report

I read carefully the authors' answer to my comments and concluded that no important changes are introduced to the manuscript, thus, unfortunately, I don't find basis to review it again (I uphold my previous advice), maybe other researcher can find some good sides of the work.

Author Response

thank you for your consideration